# Long-Short-Term-Memory-Based Deep Stacked Sequence-to-Sequence Autoencoder for Health Prediction of Industrial Workers in Closed Environments Based on Wearable Devices

**DOI:** 10.3390/s23187874

**Published:** 2023-09-14

**Authors:** Weidong Xu, Jingke He, Weihua Li, Yi He, Haiyang Wan, Wu Qin, Zhuyun Chen

**Affiliations:** 1School of Mechanical and Automotive Engineering, South China University of Technology, Guangzhou 510641, China; xuweidong@cimc.com (W.X.); menjkhe@mail.scut.edu.cn (J.H.); whlee@scut.edu.cn (W.L.); heyi@minotech.cn (Y.H.); 2Pazhou Lab, Guangzhou 510005, China; 3Future Tech, South China University of Technology, Guangzhou 510640, China; wanhy@pcl.ac.cn; 4Department of Mathematics and Theories, Peng Cheng Laboratory, Shenzhen 518000, China; 5School of Mechatronics and Vehicle Engineering, East China Jiaotong University, Nanchang 330013, China; qw@ecjtu.edu.cn

**Keywords:** health condition, workers, outlier detection, machine learning, LSTM

## Abstract

To reduce the risks and challenges faced by frontline workers in confined workspaces, accurate real-time health monitoring of their vital signs is essential for improving safety and productivity and preventing accidents. Machine-learning-based data-driven methods have shown promise in extracting valuable information from complex monitoring data. However, practical industrial settings still struggle with the data collection difficulties and low prediction accuracy of machine learning models due to the complex work environment. To tackle these challenges, a novel approach called a long short-term memory (LSTM)-based deep stacked sequence-to-sequence autoencoder is proposed for predicting the health status of workers in confined spaces. The first step involves implementing a wireless data acquisition system using edge-cloud platforms. Smart wearable devices are used to collect data from multiple sources, like temperature, heart rate, and pressure. These comprehensive data provide insights into the workers’ health status within the closed space of a manufacturing factory. Next, a hybrid model combining deep learning and support vector machine (SVM) is constructed for anomaly detection. The LSTM-based deep stacked sequence-to-sequence autoencoder is specifically designed to learn deep discriminative features from the time-series data by reconstructing the input data and thus generating fused deep features. These features are then fed into a one-class SVM, enabling accurate recognition of workers’ health status. The effectiveness and superiority of the proposed approach are demonstrated through comparisons with other existing approaches.

## 1. Introduction

In recent times, wearable devices have gained a significant position in the consumer electronics market. This can be attributed to the launch of head-mounted displays by Google Inc. [1]. Wearable devices are now considered a new means of addressing the needs of various industries [2]. For instance, the construction industry has explored the use of wearable devices for health and safety management of construction workers through proximity detection and unhealthy behaviors [3]. Some hospitals are promoting the use of wearable devices to track the habits of hospital workers and improve corporate wellness among workers [4]. Similarly, the coal mining industry has adopted a sweat monitoring wearable architecture to enable the observation of critical indicators of performance as well as the stress of mineworkers [5]. Additionally, medical professionals also utilize several wearable devices, including fitness trackers, to acquire physiological, behavioral, and contextual data for the diagnosis, treatment, and management of chronic diseases [6].

In the equipment manufacturing industry, limited space is a prevalent working environment that presents various safety risks and challenges. The confined nature of these spaces increases the likelihood of accidents, particularly larger-scale incidents that can have severe consequences. Furthermore, enclosed spaces that experience extremely high temperatures pose an even higher risk, as they can lead to heat stroke and related health issues among personnel. Given the potential dangers associated with working in confined spaces, it is essential to develop proactive measures to mitigate risks and ensure the safety of workers [7]. One crucial step in this direction is the application of machine learning techniques to predict abnormal vital signs among personnel working in such environments. By leveraging these technologies, it becomes possible to monitor vital signs in real-time and identify potential health issues before they escalate into critical situations [8,9].

In recent decades, traditional machine-learning-based abnormal detection methods have shown promise in monitoring and predicting the health conditions of humans [10,11,12]. These methods utilize machine learning algorithms such as k-nearest neighbor and support vector machines to analyze sensor data, physiological signals, or other relevant features collected from workers to identify abnormal patterns or anomalies that may indicate potential health issues or risks. For example, Venkatesan et al. [13] proposed a novel least mean square (LMS) algorithm for ECG signal preprocessing and further adopted the KNN classifier for abnormal detection, where an adaptive filter with the delayed-error-normalized LMS algorithm is utilized to attain a high speed and low latency design from ECGs. Gogoi et al. [14] proposed a breast abnormal detection method through statistical feature analysis using infrared thermograms. Šabić et al. [15] developed a machine-learning-based healthcare and anomaly detection method to predict anomalies in heart rate data, which can enhance patient care while also reducing healthcare workers’ cognitive load. Their findings demonstrated the advantages of the local outlier factor (LOF) in the task of heart rate prediction. Jalalifar et al. [16] proposed a brain tumor segmentation framework based on outlier detection using one-class SVMs. Venkatesan et al. [17] adopted an SVM classifier for abnormal detection of heart-related problems from ECG signals. Hung et al. [18] used RFID-based sensor networks to collect the elderly’s daily activities and fed the data into a hidden Markov model (HMM) and SVMs to estimate whether the elderly’s behavior is abnormal or not. Narangifard et al. [19] developed an early diagnosis of coronary artery disease through SVM, decision tree algorithms, and ensemble methods, which obtained a high recognition accuracy. Baldomero-Naranjo et al. [20] developed a robust SVM-based approach with feature selection and outlier detection for classification problems.

It can be seen that the use of machine learning algorithms can enable the analysis of vast amounts of data collected from sensors, providing valuable insights into the physiological well-being of workers in confined spaces [21]. By training models to recognize patterns indicative of abnormal vital signs, early warning systems can be developed, alerting both the workers themselves and supervisors to the need for immediate intervention [22,23]. However, traditional machine learning methods often require manual feature engineering whose process requires a deep understanding of the data and the underlying domain to identify the most informative features. It often involves pre-processing steps, such as normalization or dimensionality reduction, as well as the selection or creation of appropriate features that can effectively capture the relevant information for the task at hand. This approach can be a labor-intensive process, requiring significant time and effort [24]. As datasets grow in size and complexity, those manual approaches become even more challenging, potentially limiting the scalability and performance of traditional machine learning methods.

Recently, deep learning models have emerged as a powerful approach that eliminates the need for extensive feature engineering by automatically learning hierarchical representations from raw data [25]. The method presents advantages in comparison with signal processing methods and traditional machine learning techniques in different areas [26,27,28]. Deep neural networks, a popular type of deep learning model, have demonstrated impressive performance in addressing complex tasks like image recognition, natural language processing, and condition monitoring of assets [29,30,31]. Their ability to learn intricate patterns and representations allows them to outperform traditional machine learning methods in terms of accuracy and generalization. In the field of outlier detection in human health, deep learning methods have shown promise in addressing some challenges [32]. Karpinski et al. [33] adopted autoencoder neural networks for outlier correction in ECG-based biometric identification by using the powerful self-learning ability of the autoencoder for correcting corrupted segments. Chang et al. [34] proposed a computer-aided detection approach based on a convolutional neural network for classifying and localizing calcifications and masses in mammogram images to reduce the cost and workload of radiologists. Thomas et al. [35] constructed a hybrid outlier detection in healthcare datasets using DNN and one class-SVM. Shaban et al. [36] combined the advantages of the fuzzy inference engine and a deep neural network for detecting COVID-19 in patients, where the results were validated statistically using the Wilcoxon signed rank test and Friedman test. Verma et al. [37] used a deep convolutional neural network as a basic feature extractor for an input image and further adopted a traditional SVM to classify the image as either normal or anomalous.

From the results, it can be seen that deep learning techniques have gradually been used for outlier detection in monitoring patient health conditions and other areas. However, there is limited research focused on the health monitoring and prediction of industrial workers operating in closed environments. Several challenges need to be addressed in this area. (1) Unlike patient health monitoring, collecting health monitoring data for workers in closed situations presents challenges due to workers involving rapid movements or physical exertion and their use of protective equipment, making it difficult to install sensors. Additionally, workers in these environments may face harsh conditions like extreme temperatures, high noise levels, or chemical exposure, which further complicates data collection efforts. The physical nature of certain work environments in closed manufacturing plants can render traditional data collection methods impractical. (2) Obtaining outlier data in closed situations is indeed difficult due to the health condition of workers. In these environments, workers often operate under strict health and safety regulations. If workers experience health-related issues, they are usually required to be immediately removed from the work situation and provided with rest or medical attention. In addition, workers may have different body conditions, health histories, and susceptibilities to certain health issues, and it is difficult to obtain a comprehensive range of outlier data that cover all workers. (3) Current deep-learning-based methods for outlier detection in patient data have obtained satisfactory performance in some tasks, but rely on a large amount of available normal data, and even partial outlier data, for training the models. This requirement can hinder the practical application of these models, especially in situations where obtaining a sufficient amount of normal data and outlier data may be challenging or time-consuming.

To address these challenges, a new deep learning technique for worker health assessments in a closed environment is proposed. A wearable device called a ZJ smartwatch is developed for data collection. In this study, the cooperative company CMIC took into consideration intellectual property and information security concerns and therefore did not use the commonly used wearable device to collect health monitoring data from workers. Instead, CMIC collaborated closely with a deep partnership to develop a customized smartwatch product for the research. It is equipped with high-quality sensors that enable the collection of a wide range of health data. The device sends data to an MQTT server, which allows for easy configuration of the server address and ensures that data are transmitted securely and efficiently. Furthermore, without the use of abnormal data, a hybrid deep learning model is constructed for outlier detection of workers in closed situations. The main innovations and contributions are as follows.

(1)To ensure effective data collection despite the challenges posed by workers’ protective clothing, a novel strategy has been developed. This strategy revolves around the use of wearable intelligent bracelets as the primary method of data collection. The bracelet embedded with multi-source sensors and a Wi-Fi transfer function is specifically adapted to acquire health monitoring data from workers, even when they are wearing protective gear such as suits or equipment for tasks like operating robots. The collected data are further stored in a cloud-based platform, which can be utilized for further analysis and decision-making processes related to occupational health and safety improvements.(2)To overcome the limitations of limited data availability, a cutting-edge hybrid model is developed for outlier detection in workers operating in closed environments. This model combines an LSTM-based deep stacked sequence-to-sequence autoencoder-based approach with the one-class SVM, creating a powerful framework for effective anomaly detection. By leveraging the strengths of both the LSTM-based deep stacked sequence-to-sequence autoencoder and one-class SVM, our hybrid model can accurately identify outliers without relying on abnormal data. This is particularly useful in scenarios where obtaining abnormal data is challenging or costly.(3)To enhance the prediction ability, a feature-fused scheme has been implemented. This scheme incorporates an LSTM-based deep stacked sequence-to-sequence autoencoder component that accurately captures the underlying data structure and reconstructs the input sequence. The reconstructed error and hierarchical features learned from the autoencoder are then combined to form an improved feature representation, which is fed into a one-class SVM for outlier detection, enabling the reliable identification of outliers.

The remainder of this paper is organized as follows. The proposed method is presented in Section 2 and the experimental results are presented in Section 3. Finally, the conclusion is arranged in Section 4.

## 2. The Proposed Approach

### 2.1. Data Collection Strategy Based on Wearable Intelligent Bracelets

To ensure effective data collection even when workers are wearing protective clothing, a wearable-intelligent-bracelet-based data collection strategy is developed, as presented in Figure 1. These bracelets are adapted to be worn by workers and can capture various types of data, including health monitoring information.

The health monitoring bracelet is equipped with high-precision blood oxygen sensors, pressure sensors, temperature sensors, 3-axis accelerators, gyroscope sensors, etc. The blood oxygen sensor measures the oxygen saturation levels in a person’s blood. It provides information about how well oxygen is being transported throughout the body, which can indicate respiratory health and potential oxygen supply issues. The pressure sensor measures the pressure being exerted on the worker’s arm. It can help identify situations where excessive pressure is applied, potentially indicating the need for ergonomic adjustments or improved working conditions. The temperature sensor measures the body temperature of the worker. Monitoring body temperature is crucial for identifying fever or abnormal spikes that may indicate illness or heat-related stress, enabling timely intervention and prevention of health risks. The 3-axis accelerometer tracks movement and monitors physical activity levels, identifies postural changes, and detects falls or sudden movements that may pose a risk to the worker’s safety.

Workers in enclosed spaces wear the bracelet, ensuring close contact between the sensors and their arms. The signal acquisition device starts monitoring the bracelet’s data upon entering the work area and ceases when leaving. Throughout this period, the bracelet records the worker’s temperature, heart rate, blood oxygen saturation, number of steps, etc., which are further transferred into the cloud platform via a Wi-Fi connection. The cloud platform provides scalable storage capabilities that allow for the efficient storage and organization of the collected data. The data are structured and categorized based on different parameters, such as worker ID, date, and type of health metric. This organization facilitates easy retrieval and analysis of specific datasets when needed. The cloud platform supports collaborative functionalities, allowing multiple users to work together on analyzing data and developing health-monitoring-related intelligent maintenance algorithms, which can be utilized for further analysis and decision-making processes related to occupational health and safety improvements.

### 2.2. LSTM-Based Deep Stacked Sequence-To-Sequence Autoencoder for Health Prediction

The autoencoder is a form of feedforward neural network that is trained to reconstruct its input data from a compressed representation or latent space [38]. The idea behind an autoencoder is to learn the representation of the data that captures the important features or patterns while effectively reducing the dimensionality of the input. In order to improve the learning ability of the model, a novel LSTM-based deep stacked sequence-to-sequence autoencoder is proposed for health prediction of industrial workers [39].

With respect to traditional autoencoders with a fully connected architecture, the long short-term memory (LSTM)-based deep stacked sequence-to-sequence autoencoder is a type of neural network architecture that utilizes LSTM units. An LSTM autoencoder is an implementation of an autoencoder for sequence data using an encoder–decoder LSTM architecture. It is designed to encode and decode sequential data, such as time series or text data. In this architecture, multiple layers of LSTM units are stacked together to form an encoder–decoder structure. LSTM is a type of recurrent neural network (RNN) that excels at capturing temporal dependencies in sequential data. This makes LSTM-based deep stacked sequence-to-sequence autoencoders well-suited for tasks involving sequential data, such as time series analysis. In addition, LSTMs have a memory cell that allows them to retain information over longer sequences, capturing and utilizing contextual information effectively. This capability is especially beneficial when reconstructing sequential data, where context and past information are crucial. Traditional autoencoders do not possess this explicit memory component, making them less efficient at encoding and decoding sequences. The proposed LSTM-based deep stacked sequence-to-sequence autoencoder for the health prediction approach is presented in Figure 2.

Let us consider an LSTM-based deep stacked sequence-to-sequence autoencoder with a single-layer LSTM. The input sequence, denoted as X, is composed of a series of input vectors $X=(x^{\left(1\right)},\;x^{\left(2\right)},\;...,x^{\left(n\right)})$, where *n* is the sequence length. The LSTM consists of a recurrent layer in the encoding phase, a hidden state *h*, and a recurrent layer in the decoding phase. The encoder processes the input sequence and produces a compressed representation *z*, and the decoder reconstructs the original sequence from this representation.

In the encoder, each vector of a time-window of length is fed into a recurrent unit to perform the following computation:(1)$$h^{\left(t\right)}=LSTM_{Encoder}(x^{\left(t\right)},h^{\left(t-1\right)};\omega_{e})$$

The LSTM-based deep stacked sequence-to-sequence autoencoder generates an output sequence $Y=(y^{\left(1\right)},\;y^{\left(2\right)},\;...,y^{\left(n\right)})$ for a given input sequence X, where $\omega_{e}$ is the hyperparameters of the encoder model. Usually, X=Y to force the autoencoder to learn the semantic meaning of data. First, the input sequence is encoded by the LSTM encoder, and then the given fixed-size context variable C is decoded by the decoder LSTM.
(2)$$y^{\left(t\right)}=LSTM_{Decoder}(h^{\left(t\right)},y^{\left(t-1\right)};\omega_{d})$$

The encoder part of the network takes in a sequence of input data and processes it through the LSTM layers to extract meaningful features and create a compressed representation called the latent space also known as the bottleneck or latent space:

The objective of the LSTM-based deep stacked sequence-to-sequence autoencoder is to minimize the difference between the input sequence and its reconstruction. In this case, the max mean squared error (MSE) is commonly used:(3)$$Loss={\sum \left(x^{\left(t\right)}-y^{\left(t\right)}\right)}^{2}$$
where $x^{\left(t\right)}$ is the input sequence, $y^{\left(t\right)}$ is the reconstructed output, and n is the length of the sequence.

By minimizing the reconstruction loss, the LSTM-based deep stacked sequence-to-sequence autoencoder learns to capture the important features of the input sequence within a compressed representation and reconstruct the original sequence from that representation. During training, the gradient-based optimization algorithm is adopted to adjust the weights and biases of the LSTM networks and minimize the reconstruction loss.

### 2.3. Outer Detection with One-Class SVM with Fused Deep Features

The LSTM-based deep stacked sequence-to-sequence autoencoder is trained to encode the input sequences and obtain compressed representations that capture the essential characteristics of the data. These deep features are obtained from the bottleneck layer of the autoencoder. Then, the deep fused features that integrate the reconstruction error and bottleneck features are then used to train a one-class SVM model, which learns to classify the majority of the data as “normal” and identify any potential outliers. This SVM is trained exclusively on data from “normal” instances, as it is assumed that outlier data are scarce or unavailable during the training phase. After the one-class SVM model is trained, it can be used to detect outliers in unseen data. The optimization problem for one-class SVM can be formulated as follows:(4)$$Minimize:\;V\left(r\right)-C\;*\;\sum_{i=1}^{m}\zeta_{i}\;$$
(5)$$Subject\;to:\;\|x_{i}-o{\|}_{2}\;\leq \;r+\zeta_{i},\;i=1,2,3,...,m$$
where $V\left(r\right)$ is the volume of the hypersphere with radius r; o is the center of the hypersphere, which is a linear combination of support vectors; r is the radius of the hypersphere; $\zeta_{i}$ is a penalty factor for each training data point $x_{i}$, representing the slack variable; and C is a penalty coefficient that determines the trade-off between maximizing the volume and minimizing the influence of outlier data.

The objective of this optimization problem is to find the optimal values for o, r, and $\zeta_{i}$ that minimize the volume of the hypersphere while satisfying the constraint that all training data points are within the hypersphere. The penalty factor $\zeta_{i}$ allows for some training data points to be outside the hypersphere but at the cost of increasing the objective function.

By solving this optimization problem, one-class SVM aims to find a tight hyperspherical boundary around the non-anomalous data points, effectively separating them from the anomalous data points.

### 2.4. Specific Implement Flowchart of the Proposed Approach

The specific steps of the proposed approach are as follows.

Step 1: Data collection. Based on the edge-cloud platforms, use a Wi-Fi-bracelet to collect health condition data from workers in closed situations. This includes measurements such as temperature, heart rate, and blood oxygen saturation.

Step 2: Data pre-processing. Fuse the collected data into a single vector, and then create training data using normal data while incorporating outlier samples. Normalize the data using a standard Gaussian distribution to ensure consistency.

Step 3: LSTM-based deep stacked sequence-to-sequence autoencoder training. Train an LSTM-based deep stacked sequence-to-sequence autoencoder model using the multi-source data collected in the previous step. This model learns to extract meaningful features from the input sequences.

Step 4: Feature extraction. Extract the deep features from the trained LSTM-based deep stacked sequence-to-sequence autoencoder. These features capture essential characteristics of the input data.

Step 5: Outlier detection with one-class SVM. Feed the extracted features into a one-class SVM to detect outliers. The SVM is trained explicitly on normal data and can identify instances that deviate significantly from the learned normal patterns.

Step 6: Model testing. Evaluate the trained model using testing data. Feed the testing data into the model and assess its performance using metrics such as testing accuracy and F-score. These measures provide insights into the model’s ability to accurately detect outlier samples.

## 3. Experimental Analysis and Result Discussion

### 3.1. Dataset Descriptions

The data collection scheme was implemented in Anrui Huanke Industry, International Marine Containers (Group), to monitor the health condition of over 40 workers operating in closed spaces. These workers have various roles and responsibilities, including inspecting and maintaining robotic equipment, carrying out quality control checks and inspections, monitoring robot operations in closed spaces, and ensuring that the output or products meet the required standards. The intelligent device was used to monitor these workers and ensure the safety of operations. During the data collection stage spanning over two months, workers were selected for this study based on specific inclusion criteria. These criteria included workers who entered closed spaces for operations, had a work duration exceeding 10 min, and were above 18 years of age. For the wearable intelligent device, the sampling rate was about 8 Hz, and the variables including blood oxygen, diastolic pressure, systolic pressure, heart rate, and body temperature ware acquired for comprehensive evaluation of the health conditions of workers.

To identify individuals with abnormal vital signs, we implemented a comprehensive health assessment process. This involved conducting physical check-ups, medical history reviews, and possibly additional tests if required. The purpose was to ensure that the workers participating in the data collection were in good health and capable of providing accurate information. Certain criteria were used for obtaining the corresponding labeled data for algorithm validation based on engineering knowledge and expert experience. Workers with temperatures categorized as either 0 °C to <35 °C or ≥37.5 °C were to be regarded as having abnormal temperature readings. Those workers with heart rates falling into the range of 0 beats per minute to <60 beats per minute or ≥150 beats per minute were considered to have abnormal heart rates. Additionally, workers with blood oxygen saturation levels between 0% and <88% were considered as having abnormal blood oxygen levels. Conversely, the normal group consisted of 26 workers who exhibited normal vital signs. Their temperatures ranged from 35 °C to <37.5 °C, heart rates were between 60 beats per minute and <150 beats per minute, and blood oxygen saturation levels ranged from 88% to ≤100%. Within the study population, more than 15 cases showed abnormal vital signs, resulting in the formation of an abnormal group. In addition, before commencing the data collection process, we obtained informed consent from all participants, clearly explaining the purpose of this study, the data collection procedures, and their rights as participants. Furthermore, in order to prioritize the privacy and confidentiality of worker data, a comprehensive measure was taken to anonymize the health monitoring data of all workers. This important step involved removing any personally identifiable information and replacing it with unique identifiers or aggregated data points. By employing robust anonymization techniques, such as data encryption and pseudonymization, the sensitive health information of workers was protected from unauthorized access or disclosure.

To ensure data quality, certain measures were taken to exclude invalid or false readings. For instance, records indicating a blood pressure of 0 mmHg were flagged as they were deemed unreliable. Data with missing values, such as temperature or heart rate records during the work period, were also excluded from the analysis. Moreover, logically inconsistent data were removed, such as instances where a recorded temperature of 25 °C contradicted an actual temperature check of 36.6 °C. Finally, to better evaluate the effectiveness of the proposed method, the time series including the variables of blood oxygen, diastolic pressure, systolic pressure, heart rate, and body temperature with 5 s sampling were used for one sample. After removing the errors and redundant data, there were 2040 samples, i.e., acquired for algorithm verification. Each sample had a dimension of 200. In total, there were 408,000 data points with more than 850 min acquired for the performance test. The specific descriptions of the adopted samples in the experiments can be seen in Table 1. It should be noted that the normal data refer to the health condition data.

### 3.2. Baseline Methods

To demonstrate the advantages of the proposed method, three different techniques are used for algorithm comparisons: local outlier factor, one-class SVM, and deep autoencoder. Detailed below are descriptions of these compared methods.

Method 1 (Local Outlier Factor, LOF) [15]: The local outlier factor is an algorithm used for outlier detection in a dataset. In LOF, the k-NN approach is used to estimate the local density around each data point. For a given data point, its *k* nearest neighbors are identified based on a distance metric such as Euclidean distance. The local reachability density (LRD) of the data point is then computed by considering the inverse of the average distance between the data point and its k nearest neighbors. It considers the local deviation of a data point from its neighbors to identify potential outliers. By calculating a score for each point, the algorithm can determine the degree of abnormality compared to its local neighborhood. A higher LOF score indicates a higher likelihood of being an outlier. The algorithm takes into account density variations in different regions of the dataset and is capable of handling non-linear relationships between variables. As a result, LOF is a flexible and effective method for detecting local anomalies in various applications. The algorithm has been successfully used for heart rate anomaly detection.

Method 2 (Isolation Forest): Isolation Forest, an ensemble of decision trees, is an unsupervised machine learning algorithm used for anomaly detection of health condition monitoring. It is particularly effective in identifying outliers or anomalies within a large-scale dataset. The algorithm works by isolating instances through a process of randomized partitioning. It randomly selects a feature and then randomly selects a split value within the range of that feature. By recursively partitioning the data based on these random splits, the algorithm creates a set of isolation trees. The anomaly score of each instance is calculated based on the number of normal data points of health conditions required to isolate it. Instances that can be isolated with few abnormal data points of health conditions are considered more anomalous, while instances requiring more partitions are considered less anomalous or normal.

Method 3 (one-class SVM with linear kernel) [20]: One-class SVM is a support vector machine algorithm that learns a decision boundary to separate normal instances from outliers while maximally covering the normal instances. It is specifically designed for outlier detection and has been widely used in anomaly detection scenarios. It is a powerful algorithm that learns a decision boundary around the normal instances, making it suitable for our purposes in comparison with traditional supervised machine learning methods such as logistic regression and naïve Bayes. The linear kernel is used to create a linear decision boundary in the feature space. One-class SVM with a linear kernel is suitable for linearly separable data and provides a robust way of detecting outliers.

Method 4 (one-class SVM with RBF kernel) [20]: The RBF kernel is a popular choice for SVMs as it allows for nonlinear decision boundaries by mapping the input data into a higher-dimensional feature space. The one-class SVM with the RBF kernel aims to find a hypersphere in a feature space that encompasses the majority of the training data, representing the normal class. Any data points outside the hypersphere are considered anomalies. The one-class SVM has been adopted for outlier detection of a leukemia dataset.

Method 5 (one-class SVM with poly kernel) [20]: In addition to the linear kernel, one-class SVM with non-polynomial kernels allows for more complex decision boundaries, enabling the detection of non-linear patterns in the data. One-class SVM with polynomial and RBF kernels is particularly useful when dealing with data that exhibit non-linear relationships between features.

Method 6 (Handcrafted features and one-class SVM): To assess the efficacy of the hand-crafted features, a comprehensive set of ten statistical features is utilized. These features include the mean, root mean square, variance, root mean square amplitude, maximum absolute value, skewness, kurtosis, peak-to-peak value, maximum value, and minimum value. These features are extracted from the raw health monitoring data, enabling a thorough characterization of the dataset. Subsequently, the extracted features are employed as input for the one-class SVM algorithm, which accurately discerns between normal and abnormal samples. By leveraging the approach, we can effectively identify and classify anomalies within the health monitoring data, contributing to enhanced detection and diagnosis capabilities.

Method 7 (Autoencoder neural networks) [33]: The deep autoencoder is an unsupervised neural network architecture that consists of an encoder and a decoder. It plays a crucial role in learning deep and compressed representations of input data. The encoder attempts to learn a lower-dimensional representation of the input data by mapping it to a compressed latent space, while the decoder reconstructs the original input from the latent space. By training the autoencoder on normal instances, it learns to reconstruct them accurately, capturing their underlying patterns and structures. Unlike the LSTM-based autoencoder, reconstructing sequential data to retain long-term dependencies by LSTM structure, for the autoencoder construction, both the encoder module and decoder module include two-layer fully connected layers. The architecture is [200-50-30-50-200], which means that the hidden nodes of the first and second layers in the encoder stage are 200 and 50, respectively. The node of the bottleneck layer is 30. In addition, the nodes of the first and second layer in the decoder stage are 50 and 200, respectively. The activation function is ReLU. During the training procedure, the Adam algorithm is adopted for parameter optimization. The batch size is 30, and the number of epochs is set to 300 for full model training. During the outlier detection, the reconstruction error is used for the input of one-class SVM, which is employed to create a boundary separating the normal instances from the outliers. The one-class SVM learns to define a hypersphere in the latent space that encapsulates the normal instances. The instances lying outside the hypersphere are classified as anomalies.

Method 8 (Proposed without deep feature fusion, called Proposed_wo)/Method 9 (Proposed with deep feature fusion, named Proposed). In proposed method 6, the reconstruction error without the fusion of deep features is directly used for the input of one-class SVM. For model construction, both the encoder module and decoder model include two-layer LSTMs, which are stacked together for deep feature learning. The number of embedding dimensions and hidden nodes is set to 64 and 50, respectively. The Adam optimization algorithm is adopted for model training, and 30 epochs are implemented for full training.

### 3.3. Evaluation Indexes

In the experiments, the performance of various models using metrics including Accuracy and F-score are adopted for comprehensive evaluation. When dealing with binary classification problems, specific errors are defined. False positive (*FP*) refers to the situation where a normal instance is incorrectly classified as anomalous. False negative (*FN*) occurs when an anomaly instance is wrongly labeled as normal. Conversely, true positive (*TP*) and true negative (*TN*) represent the correct identification of anomaly and normal instances, respectively. These metrics can be evaluated as follows [40].
(6)Precision=TP/(TP+FP)
(7)Recall=TP/(TP+FN)
(8)Accuracy=(TP+TN)/(TP+TN+FP+FN)
(9)Fscore=2∗(Precision∗Recall)/(Precision+Recall)

### 3.4. Result Analysis

#### 3.4.1. Visualization of the Learned Features

To better visualize what the deep network has learned from the input data, the raw data and deep features extracted using the proposed methods were visualized using PCA reduction techniques [41]. The raw high-dimensional data and deep features were reduced to 2D for feature visualization. In Figure 3, normal classes are represented in a cycle shape, while outlier classes are represented in a cross shape. The results show that the raw data display significant distribution discrepancies even within the same class. For example, the normal class in the raw data scatters across different regions, indicating a lack of adherence to a consistent data distribution. Similarly, the outlier data cluster in different regions and overlap extensively with the normal data, making it difficult to learn a decision boundary for detecting outlier samples. In contrast, the proposed method achieves good cluster performance in both normal data and outlier data. Specifically, the normal class is tightly clustered together, while the outlier data are located in distinct regions. This demonstrates that the deep features possess strong discriminative abilities, which contribute to improved detection performance.

In addition, to provide a quantitative emulation of the visualization results, three novel evaluation indexes are constructed. Specifically, for the input data X, which is derived from the analyzed dataset, where M represents the total number of samples, we first define two scatter matrices: the between-class matrix ($S_{b}$) and the within-class matrix ($S_{w}$). The between-class matrix ($S_{b}$) is a square matrix that represents the scatter or variation between different classes in a dataset. It quantifies the differences between the means of different classes in a multiclass classification problem [42,43]. Here, the weight used is the number of samples in each class. Similarly, the within-class matrix ($S_{w}$) is a square matrix that represents the scatter or variation within each class in a dataset. It quantifies the differences between individual samples and their respective class means. The quality of learned features can be evaluated using three measure indexes that combine the scatter degree and concentration degree.
(10)$$J_{1}=\frac{\left|S_{b}\right|}{\left|S_{w}\right|}$$
(11)$$J_{2}=\frac{\mathrm{Tr}[S_{b}]}{\mathrm{Tr}[S_{w}]}$$
(12)$$J_{3}=\frac{\left|S_{b}+S_{w}\right|}{\left|S_{w}\right|}$$
where |A| represents the 2-norm (also known as the magnitude or Euclidean norm) of matrix A, while “Tr(A)” denotes the trace of matrix A. A larger scatter degree indicates that the features from different classes can be more easily separated, implying better class discrimination. On the other hand, a smaller concentration degree suggests that features belonging to the same class are more tightly clustered together, indicating stronger intraclass similarity. Therefore, a larger evaluation value signifies that the diagnosis model is more effective in reducing the discrepancy in data distribution, indicating improved performance.

The results presented in Table 2 show that the raw data have the lowest values in all the evaluation indexes. This suggests that there is a poor separation between classes and a high degree of overlap among different classes when considering direct visualization analysis. In general, the three evaluation indexes effectively capture the distribution discrepancy of features extracted from different methods. Specifically, the proposed method achieves the best results, indicating a superior feature learning capability and improved performance in domain adaptation diagnosis.

#### 3.4.2. Effectiveness of the Proposed Approach on Limited Training Samples

To assess the effectiveness of the proposed approach on limited training samples, a dataset consisting of only 50 samples from the health condition data of workers in closed situations is used for model training. Specifically, the leave-one-out method is adopted for selecting the training data. The method ensures that each person’s data are used for testing once, while the remaining subjects’ data are utilized for training. Five repetitive experiments are conducted and the performance of the model is evaluated using average accuracy and F1-score as metrics. Average accuracy is a metric that measures the overall correctness of the outlier detection model. It calculates the average proportion of correctly classified instances, considering both normal and outlier instances. F-score is a metric that balances the model’s precision and recall. It provides a single measure that combines both of these aspects, making it suitable for assessing the overall effectiveness of an outlier detection model. Both average accuracy and F1-score provide valuable insights into the performance of an outlier detection model.

From the results presented in Table 3, it is evident that the compared outlier detection methods, including Method 1 (local outlier factor), Method 2 (isolation forest), Method 3, Method 4 and Method 5 (SVM with different kernels), Method 6 (handcrafted features and one-class SVM), and Method 7 (deep autoencoder), achieve a similar recognition testing accuracy of nearly 70%. This indicates their effectiveness in learning and detecting outliers. When considering the F-score, which takes into account both precision and recall, it becomes evident that Method 6, which utilizes handcrafted features in conjunction with the one-class SVM, yields lower accuracy when compared to the other methods. This discrepancy can be attributed to the fact that the extracted statistical features are highly susceptible to fluctuations and variations in the health monitoring data, particularly due to the inherent nature of the work environment where different workers exhibit significant fluctuations in their health conditions. As a result, the performance of Method 6 is compromised, highlighting the challenges associated with relying solely on handcrafted features for anomaly detection in such dynamic and closed environments. On the other hand, Method 7 (deep autoencoder) demonstrates better performance among the compared methods, suggesting that the use of deep architecture helps in learning better feature representations and obtaining improved results compared to shallow methods. In contrast, the proposed method, consisting of both Method 8 (Proposed without fused features, Proposed_wc) and Method 9 (Proposed without fused features, Proposed), achieves the best results in terms of both accuracy and F-score. This highlights its superior classification performance over the other compared methods. The proposed method’s ability to perform well suggests that it incorporates innovative techniques or features that enhance its outlier detection capabilities.

#### 3.4.3. Effects of the Hidden Nodes on Recognition Performance

In this part, the effects of the hidden nodes of the proposed LSTM-based deep stacked sequence-to-sequence autoencoder on the final classification performance are investigated. The number of hidden nodes is varied in the range from 30 to 150, and the average accuracy and F-score are presented in Figure 4. It can be observed that the proposed model is less sensitive to changes in the number of hidden nodes. Regardless of the number of nodes, all models achieve a testing accuracy of more than 71% and an F-score of over 78%. This indicates the model’s strong learning performance and ability to adapt to different configurations of hidden nodes. However, it should be noted that using a large number of nodes or a small number of nodes can deteriorate the recognition results. Therefore, finding an optimal number of nodes is crucial for achieving the best performance. Upon examining the results, it is found that the best results are obtained when the number of nodes equals 50. The particular configuration is selected as the optimal parameter for the proposed model. This implies that a moderate number of hidden nodes strikes a balance between representation capacity and computational efficiency, leading to improved classification performance.

#### 3.4.4. Effects of the Proposed Approach on a Large Number of Training Samples

In this section, additional experiments were conducted to further validate the superiority of the proposed approach over other techniques. The dataset used in these experiments consisted of 100 training samples from the health condition data, while the remaining health data and outlier data were utilized for testing. The recognition performance of the model is presented in Figure 5. Upon analyzing the results, it is evident that as the number of training samples increases, both models achieve a higher detection accuracy. This suggests that a larger number of samples are beneficial in enriching the diagnostic knowledge for learning tasks of all the data-driven approaches. Furthermore, Method 7 (deep autoencoder) performs better compared to the other compared methods, indicating its good performance in extracting discriminative features through its deep structure. However, despite the strong performance of the autoencoder model, the proposed approach still obtains the best results, demonstrating its effectiveness and superiority over the other techniques. These findings reinforce the notion that the proposed approach incorporates innovative techniques or features that contribute to its exceptional performance in outlier detection and classification tasks.

## 4. Discussion

Some interesting findings can be drawn from the analysis above.

(1)Due to noise interference and bias, as well as high data dimensionality, the original data easily obscure discriminative information, posing a significant challenge for existing intelligent system algorithms.(2)Traditional machine learning methods have a computational advantage in constructing shallow machine learning models that learn different patterns in the input. However, they have limitations when dealing with high-dimensional nonlinear health monitoring data, and their diagnostic effectiveness may heavily rely on the quality of feature extraction.(3)Compared to traditional machine learning models, deep learning methods have a core advantage in directly learning discriminative representations from traditional input data. These representations are beneficial for providing effective feature representations for subsequent anomaly detection tasks.(4)The proposed method has clear advantages compared to traditional machine methods and also achieves good performance compared to deep learning methods (autoencoder). This may be attributed to the LSTM network’s ability to effectively learn temporal dependencies from the original data, thereby contributing to better results.

## 5. Conclusions

In this paper, a novel approach called an LSTM-based deep stacked sequence-to-sequence autoencoder is proposed for predicting the health status of workers in confined spaces. By leveraging the benefits of the LSTM-based deep stacked sequence-to-sequence autoencoder and integrating it with hybrid deep learning and a one-class SVM architecture, the proposed approach aims to enhance the prediction accuracy of workers’ health status in confined spaces. Ultimately, it contributes to improving safety and productivity in challenging working environments. The effectiveness and superiority of the proposed approach are validated in comparison with other traditional machine learning techniques and deep-learning-method-based outlier detection methods.

Despite the promising performance of the proposed scheme, there are still some issues that need to be addressed. One concern pertains to the limited scope of data collection, as only five different sensor data sources were selected for this study. However, considering the correlation between various physiological parameters and the health status of the human body, it is crucial to explore and determine appropriate monitoring parameters that account for the interdependencies and noise coupling among these parameters. Furthermore, the hyperparameters of the proposed method were determined through human selection. To enhance the intelligence of the system, it would be worthwhile to investigate the integration of auto-machine learning techniques. This represents an important area for further research and improvement.

## Figures and Tables

**Figure 1 sensors-23-07874-f001:**
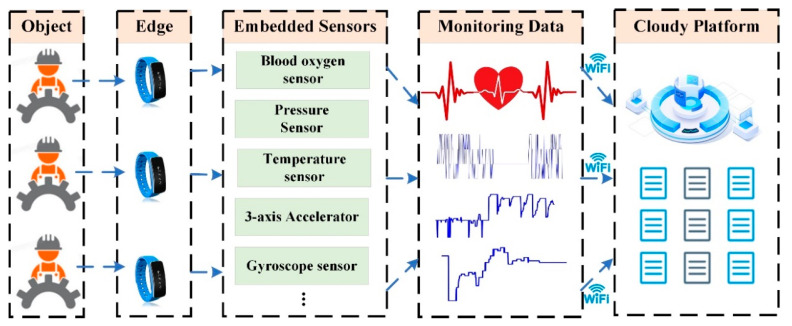
The data collection scheme with wearable intelligent device.

**Figure 2 sensors-23-07874-f002:**
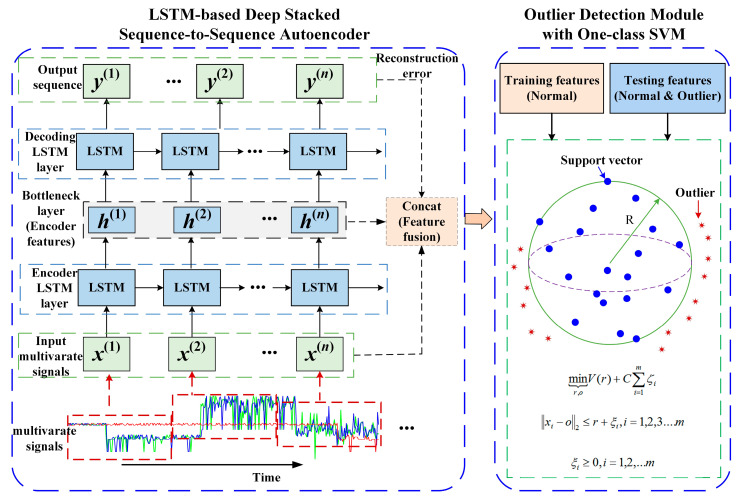
LSTM-based deep stacked sequence-to-sequence autoencoder for health prediction.

**Figure 3 sensors-23-07874-f003:**
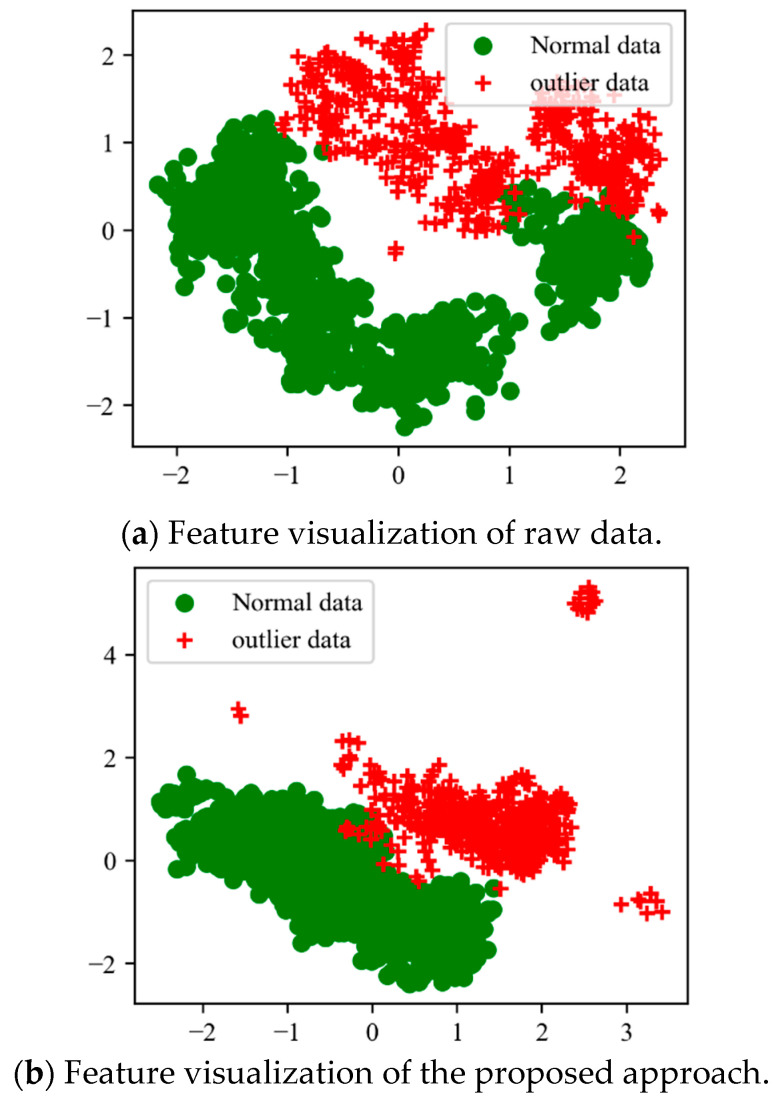
Visualization of the learned features.

**Figure 4 sensors-23-07874-f004:**
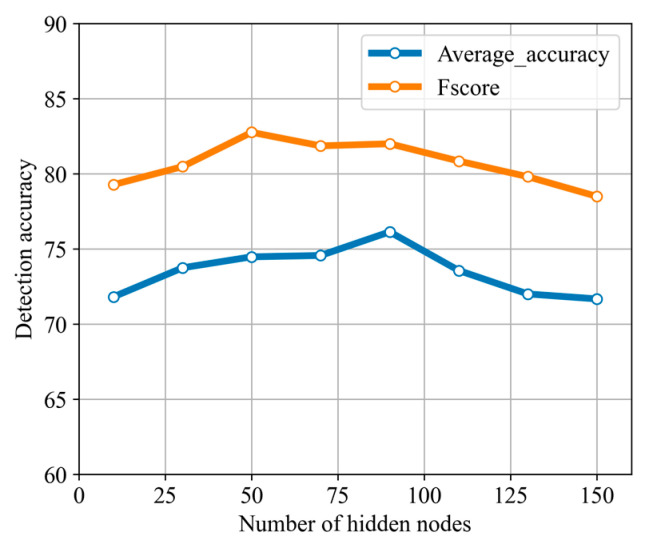
Effects of the number of hidden nodes on detection results.

**Figure 5 sensors-23-07874-f005:**
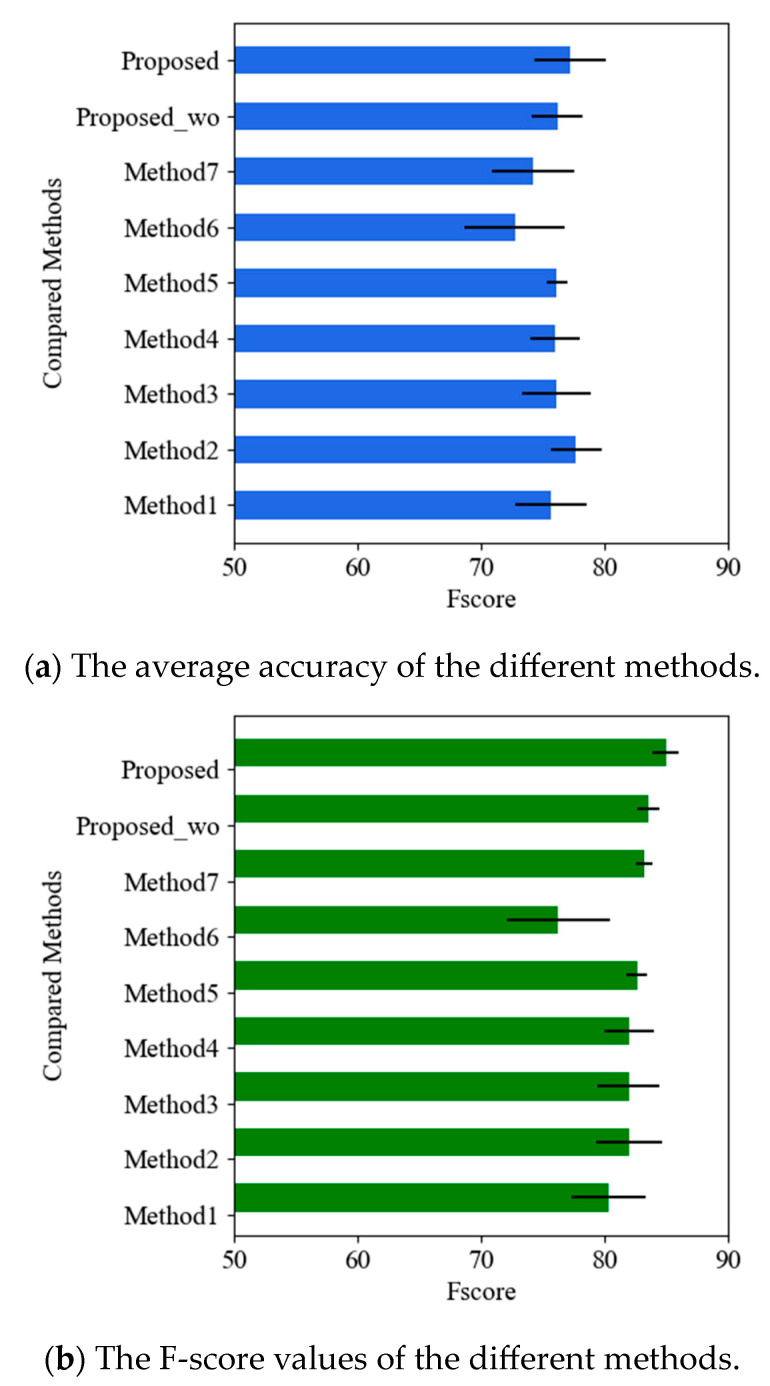
Detection results among different methods under a large number of training samples.

**Table 1 sensors-23-07874-t001:** Descriptions of the constructed datasets.

Datasets	Number of Samples	Acquired Varies
Normal data	1500 samples with a dimension of 200 × 1	Blood oxygen; Diastolic pressure; Systolic pressure; Heart rate; Body temperature
Abnormal data	540 samples with a dimension of 200 × 1	

**Table 2 sensors-23-07874-t002:** The quantitate evaluation results of two methods with different learned features.

Different Features	Raw Data	Extracted Deep Features
J1	1.39	2.67
J2	0.50	1.04
J3	0.41	0.86

**Table 3 sensors-23-07874-t003:** The detection results among different methods under limited training samples.

Categories	Methods	Average Accuracy	F-Score
Shallow learning	Method 1	71.51 ± 1.98	77.36 ± 2.04
	Method 2	73.24 ± 3.43	77.41 ± 3.43
	Method 3	71.97 ± 3.23	78.05 ± 2.70
	Method 4	72.58 ± 3.01	78.54 ± 3.18
	Method 5	72.49 ± 2.92	78.49 ± 2.62
Deep learning	Method 6	71.49 ± 4.62	79.87 ± 3.39
	Method 7	69.25 ± 4.91	72.50 ± 5.75
	Proposed_wo	75.04 ± 2.55	81.52 ± 2.28
	Proposed	74.47 ± 4.54	82.75 ± 3.71

## Data Availability

The data presented in this study are available on request from the corresponding author.

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
