# Peer review of "Long-Short-Term-Memory-Based Deep Stacked Sequence-to-Sequence Autoencoder for Health Prediction of Industrial Workers in Closed Environments Based on Wearable Devices"

_sensors, 2023, doi:10.3390/s23187874_

Round 1

Reviewer 1 Report

The topic of this article is interesting and important: how to monitor health conditions of industrial workers. However, the paper has several problems:

1. Name the used wearable device

2. Introduction could be reorganized. Tell first about industrial studies and then more to other related works

3. How many minutes of data you have?

4. Explain method 5

5. In the experiments you talk about health condition data, does this mean Normal data (Table 1)?

6. Features used in the experiments are extracted using the proposed deep learning method, compare the results of this to hand picked features

6. The biggest problem of the work is that it is not explained how the training data is selected for the experiments. Have you randomly chosen these 50 samples (or 100)? In this type of studies, leave-one-out method needs to be used to select training data. This means that in turn one persons data is used for testing and training data is selected from the remaining subjects data. Random selection leads to over-fitting as then you may have same persons data in training and test sets.

7. When you select for instance 50 samples to train the model, run the model several times with different datasets. Show results form these runs

8. section 3.4.4. could be before other results

9. Discussion missing

Some typos.

Author Response

The authors would like to express their sincere appreciation to the editor and reviewers for their constructive comments and efforts in helping us to further improve the quality and presentation of the manuscript. The authors have carefully revised and proofread the whole manuscript, and hope that the revised manuscript has addressed the issues raised by the reviewers and provides a clearer and more informative description of our work. The revision is marked in blue in the revised manuscript.

Please see our point-by-point responses below.

Reviewer 2 Report

The authors proposed a LSTM-Based Deep Stacked Sequence-To-Sequence Autoencoder  method on the collected multi-source data to predict the workers health status. I here have some concerns about the submitted paper.

1. Please recheck the writing to correct the unexpected typos, like brackets on line 20,  a worker they on line 113, etc.

2. This looks a bit overused in the manuscript, which usually leads to the issue of unclear untecedent. Please consider rewriting the sentences to remove the unclear reference.

3. The proposed method used deep networks to learn features for SVM. Please explain why you used one-class SVM as binary classifier instead of Logistic Regression, k-Nearest Neighbors, Decision Trees, Naive Bayes. Or why did not you just use fully-collected layers with softmax as the classifier for simplicity?

4. The compared methods should be called baseline methods?

5.I would like to know why you just compared your methods with the SVM-based methods, how about the methods mentioned in comment 3.

6. How did you know whether a worker is healthy or not when you collected the data? Have you provided like the ethical agreements for your data collection.   

Language editing is needed for the manuscript。

Author Response

(The authors gave the same response as above.)

Round 2

Reviewer 1 Report

Hi, this looks much better now!

Little modifications are still needed:

You still have not named the device you are using. In order in improve the repeteability of your study, tell the brand and the model of the device you are using.

Line 408: Method 7 (put the name of the method here in parenthesis as you have done in other cases as well) [33]:

-

Author Response

The authors would like to express their sincere appreciation to the editor and reviewers for their constructive comments and efforts in helping us to further improve the quality and presentation of the manuscript. The authors have carefully revised and proofread the whole manuscript, and hope that the revised manuscript has addressed the issues raised by the reviewers and provides a clearer and more informative description of our work. The revision is marked in blue in the revised manuscript.

Please see our point-by-point responses in attached file "Response_to_Reviewers-sensors-2565538-R2.docx"

Thank you very much

Kind regards,

All authors

Reviewer 2 Report

The manuscript has been sufficiently improved to warrant publication in Sensors, but some typos shoulld be further corrected. 

I found at least three 'bracket' in the manuscript,which should be bracelet? Please recheck the writing.

Author Response

(The authors gave the same response as above.)
